# Knowing When to Stop: Efficient Context Processing via Latent Sufficiency Signals

**Roy Xie**[†][*]    **Junlin Wang**[†]    **Paul Rosu**[†]    **Chunyuan Deng**[‡]

**Bolun Sun**[§]    **Zihao Lin**[‖]    **Bhuwan Dhingra**[†]

[†]Duke    [‡]Rice    [§]JHU    [‖]UC Davis

## Abstract

Large language models (LLMs) process entire input contexts indiscriminately, which is inefficient when the information required to answer a query is localized within the context. We present dynamic context cutoff, a novel method enabling LLMs to self-terminate processing upon acquiring sufficient task-relevant information. Through analysis of model internals, we discover that specific attention heads inherently encode "sufficiency signals" – detectable through lightweight classifiers – that predict when critical information has been processed. This reveals a new efficiency paradigm: models' internal understanding naturally dictates processing needs rather than external compression heuristics. Comprehensive experiments across six QA datasets (up to 40K tokens) with three model families (LLaMA/Qwen/Mistral, 1B-70B) demonstrate 3.4% accuracy improvement while achieving $1.33\times$ token reduction on average. Furthermore, our method demonstrates superior performance compared to other context efficiency methods at equivalent token reduction rates. Additionally, we observe an emergent scaling phenomenon: while smaller models require probing for sufficiency detection, larger models exhibit intrinsic self-assessment capabilities through prompting. Code is available at https://github.com/ruoyuxie/when-to-stop.

## 1   Introduction

Large language models (LLMs) demonstrate remarkable capabilities across diverse tasks, yet their indiscriminate processing of entire input contexts creates inefficiencies. LLMs process every token with equal computational priority, regardless of its actual relevance to the task [25]. This brute-force approach leads to fundamental inefficiencies: models waste computation on irrelevant context while simultaneously struggling with the "lost-in-the-middle" phenomenon, where critical information becomes diluted in lengthy inputs [16, 6]. For instance, when answering a simple factual question, models may process an entire document even after gathering sufficient information in the first few sentences.

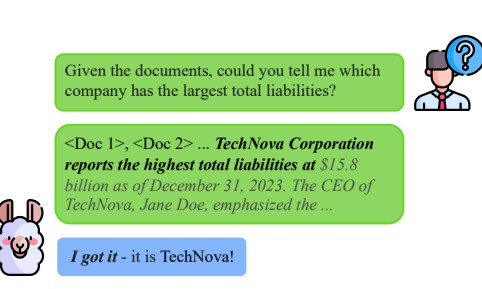

Figure 1: Our method enables language models to perform early termination by detecting sufficiency signals in key attention heads, reducing the amount of processed content while preserving performance.

The human cognitive system offers an instructive contrast. When solving problems, people dynamically assess information sufficiency – we stop processing once we gather enough evidence, ignoring redundant details [4]. On the other hand, LLMs process entire contexts even after acquiring sufficient

---

[*]ruoyu.xie@duke.edu

39th Conference on Neural Information Processing Systems (NeurIPS 2025).

information. This raises a question: "Can we enable LLMs to self-assess context sufficiency and terminate early without compromising accuracy?"

In this work, we present dynamic context cutoff, which enables LLMs to identify when they have acquired sufficient information for a task. Our key insight emerges from analysis of model internals: specific attention heads in transformer layers exhibit strong sensitivity to information sufficiency (§2.2). By monitoring these "context sufficiency heads" with lightweight classifiers, we enable models to make early stopping decisions while improving performance.

Context compression offers a promising avenue for improving inference efficiency in LLMs. However, existing compression methods typically operate by predefining a target compression rate, which introduces the risk of information loss. For instance, the LLMLingua family [8, 10, 18] employs a small language model to filter out unimportant tokens, reducing context based on a fixed compression target. These methods impose predefined compression rates, enforcing a *one-size-fits-all* reduction regardless of content complexity. Similarly, retrieval-augmented generation (RAG) methods predefine a fixed number of top-$k$ retrieved documents. Although RAG operates differently by retrieving external documents rather than compressing existing input, we include it for comprehensive comparison as it has appeared in the baselines of previous work on context compression [10, 18]. We refer to both compression-based approaches and RAG as *static* methods, as they apply uniform compression ratios (e.g., 50% compression results in every input being compressed to exactly half its length). In contrast, our method is *dynamic and context-adaptive* – different inputs receive different amounts of compression based on their information density, with actual compression determined by each input's specific content rather than a predetermined target. This approach enables models to process only the minimal context needed, expanding it only when necessary, creating a new paradigm where *efficiency emerges naturally from the model's own understanding* rather than from external compression heuristics, as demonstrated in Figure 1.

We conduct comprehensive experiments to evaluate our approach across six QA datasets (context lengths 0.5K–40K tokens) and three model families (LLaMA, Qwen, Mistral; 1B–70B parameters). We find that LLMs inherently encode context sufficiency signals in specific attention heads. Notably, our method reveals behaviors that align well with the scaling properties of modern LLMs: while smaller models (1B–8B parameters) require explicit sufficiency detection to achieve competitive efficiency, larger models (14B+) exhibit emergent self-assessment capabilities through simple prompting. Our method achieves a 3.4% average accuracy improvement with $1.33\times$ token reduction, outperforming state-of-the-art context compression methods. Our in-depth analysis explores the sensitivity to classification thresholds, the efficiency gains from different chunking strategies, and the model-specific nature of context sufficiency, providing valuable insights into context sufficiency.

## Related Work

**Efficient Context Processing in LLMs.** Improving inference efficiency in LLMs has attracted significant research attention. Existing approaches fall into two orthogonal categories: (1) methods that approximate transformer computations during inference, including speculative decoding [13], quantization [32], efficient attention mechanisms [1, 12], and KV cache optimization [37, 9, 29]; (2) methods that reduce input context length through compression [8, 10, 18, 28, 15]. These approaches are complementary – any context compression method can benefit from computational approximations. Our work focuses on context reduction rather than computational approximation. Specifically, our method aligns with hard prompt compression approaches like the LLMLingua family, which compress input at the *textual* level without requiring model retraining. We include the full suite of LLMLingua variants in our experiments. For comprehensive comparison, we also include the RAG baselines used in [10, 18], though RAG operates differently by retrieving external documents rather than compressing existing input context. Critically, unlike both LLMLingua and RAG, our method does not require any predefined compression target and dynamically adjusts context length based on the model's own understanding of the input content.

**Latent Knowledge in Model Activations.** LLMs encode task-relevant knowledge within intermediate activations [23], often more accurately than their final surface outputs [22, 14]. This latent knowledge has been leveraged for various downstream applications, including knowledge graph construction [27], reasoning correctness verification [35], hallucination detection [7], and long-context understanding [34]. We extend this line of work by interpreting model internal activation subspaces

to detect context sufficiency. Unlike prior work, our approach uniquely identifies *when the model has internally synthesized adequate information* and leverages this insight to reduce information overhead during context processing. To the best of our knowledge, we are the first to propose a dynamic context cutoff method that uses the model's internal signals to determine when to stop processing context.

## 2 Methodology

We propose dynamic context cutoff, a method that enables LLMs to identify and process only the minimal sufficient context required for a given task. Our approach leverages internal model activations to detect when enough information has been gathered, reducing token processing while improving performance, as illustrated in Figure 2.

### 2.1 Problem Formulation

Let $\mathcal{M}$ denote a pre-trained language model. Given an input textual context $\mathbf{C}$, we process $\mathbf{C}$ sequentially from left to right by partitioning it into an ordered sequence of chunks $\{\mathfrak{s}_j\}_{j=1}^m$, where each chunk $\mathfrak{s}_j$ comprises a contiguous subset of $\mathbf{C}$ (e.g., a sentence or a fixed percentage of the total tokens). These chunks form a non-overlapping covering of $\mathbf{C}$, meaning:

$$\Big\|_{j=1}^m \mathfrak{s}_j = \mathbf{C}, \quad \mathfrak{s}_i \cap \mathfrak{s}_j = \emptyset \text{ for } i \neq j$$

where $\|$ denotes concatenation. We define a sequence of cumulative contexts $\{\mathbf{C}_i\}_{i=1}^m$, where each cumulative context $\mathbf{C}_i$ consists of all chunks up to and including the $i$-th chunk:

$$\mathbf{C}_i = \mathfrak{s}_1 \,\|\, \mathfrak{s}_2 \,\|\, \ldots \,\|\, \mathfrak{s}_i, \quad 1 \leq i \leq m$$

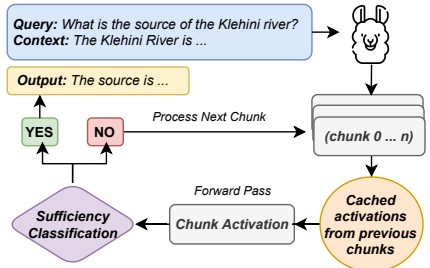

Figure 2: Our method leverages the model's internal representations to identify when sufficient information has been processed. A lightweight classifier is trained on selected attention heads to detect context sufficiency, leading to token savings while improving task performance.

By construction, these cumulative contexts satisfy the nested proper subset relationship: $\mathbf{C}_1 \subset \mathbf{C}_2 \subset \cdots \subset \mathbf{C}_m = \mathbf{C}$. Given a query $q$, the goal is to identify the smallest prefix $\mathbf{C}_k$ (where $k \leq m$) such that:

$$\mathcal{M}(q, \mathbf{C}_k) \approx \mathcal{M}(q, \mathbf{C}).$$

Here, $\mathbf{C}_k$ represents the minimal sufficient context required for the model to answer $q$ with comparable performance to using the full context $\mathbf{C}$.

At each step $i$, a sufficiency classifier $\mathcal{S}$ iteratively checks whether the current cumulative context $\mathbf{C}_i$ contains enough information, by comparing its confidence $\mathcal{S}_c(C_i)$ with a threshold $\tau$. Formally,

$$\mathcal{S}(\mathbf{C}_i) = \begin{cases} 1 & \text{if } \mathcal{S}_c(\mathbf{C}_i) \geq \tau, \\ 0 & \text{otherwise} \end{cases},$$

where $\mathcal{S}_c : \mathbb{R}^d \to [0, 1]$ is the sufficiency confidence score function, and $\tau$ is a predefined threshold. If $\mathcal{S}(\mathbf{C}_i) = 1$, processing terminates, and $\mathbf{C}_i$ is selected as the minimal sufficient context $\mathbf{C}_k$. The remaining chunks $\{\mathfrak{s}_{i+1}, \mathfrak{s}_{i+2}, \ldots, \mathfrak{s}_m\} = \mathbf{C} \setminus \mathbf{C}_k$ are ignored.

We formulate our task as a *left-to-right context processing* problem rather than searching or selecting an arbitrary subset of documents. This aligns with how LLMs naturally process text from left to right to maintain semantic coherence and continuity between chunks. Our cumulative, non-overlapping chunking approach is essential for computational efficiency: each new chunk extends the context incrementally (chunk$_1$, chunk$_{1+2}$, chunk$_{1+2+3}$, etc.), allowing us to reuse the KV cache and avoid redundant computation of previously processed tokens. Overlapping chunks would require recomputing activations for the same tokens multiple times, negating the computational efficiency gains that make this approach practical. More discussion in Appendix B.3.

## 2.2 Probing LLMs for "Context Sufficiency"

We are interested in understanding how context sufficiency is represented within the model and how it can be leveraged to improve efficiency. To do so, we probe its intermediate activations. Following prior work on neural network interpretability [14, 2], we assess whether certain attention heads encode information predictive of sufficiency. The data for probing consists of input cumulative contexts $\{\mathbf{C}_i\}_{i=1}^n$, each labeled as either sufficient ($y = 1$) or insufficient ($y = 0$).

For each $\mathbf{C}_i$, the model produces attention head activations $\{x_l^h\}$, where $x_l^h \in \mathbb{R}^D$ is the activation of the $h$-th head in the $l$-th layer. We train a lightweight binary classifier $p_\theta(x_l^h)$ on these activations to predict sufficiency: $p_\theta(x_l^h) = \sigma(\langle \theta, x_l^h \rangle)$, where $\theta \in \mathbb{R}^D$ are the parameters of the probe, and $\sigma$ denotes the sigmoid function. The dataset is split into training and validation sets (4:1 ratio) per task. We discuss more details about the data used for the probe in §A.3. Each classifier's validation F1 score determines the predictive ability of the corresponding head. This selection process is performed only once for all tasks. Figure 3 shows the F1 scores of probes for all attention heads in LLaMA3.2-1B. A subset of heads, primarily in middle layers, exhibit significantly higher predictive performance. However, the performance of the probes may vary depending on model architecture.

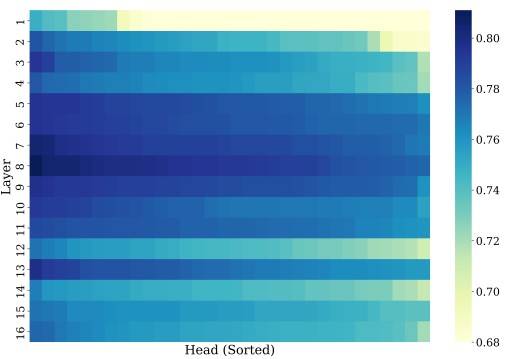

Figure 3: Validation F1 scores for linear probes across all attention heads in LLaMA3.2-1B, sorted row-wise by F1. Darker blue represents higher F1 scores. Some heads show significantly higher performance. More visualizations can be found in Figure 8.

These results suggest that the model's internal representations encode latent information about context sufficiency. We leverage this insight to identify the most informative attention heads for further processing. More details about the head selection process can be found in Appendix B.1.

## 2.3 Dynamic Context Cutoff

**Sufficiency Classification.** After identifying the top heads from the probing step, we train multiple lightweight base classifiers $\{\mathcal{S}_1, \mathcal{S}_2, \ldots, \mathcal{S}_e\}$ on these heads to form an ensemble. The ensemble is constructed using StratifiedKFold with $n = 5$ folds, with the best performing models selected based on their mean cross-validation AUC scores to form the final weighted ensemble:

$$\mathcal{S}_{\text{ensemble}}(\mathbf{C}_i) = \frac{1}{e} \sum_{j=1}^e \mathcal{S}_j(\mathbf{C}_i).$$

More details on this ensemble classifier can be found in Appendix F.1.

**Inference with Iterative Forward Passes.** During inference, the full context is processed incrementally as a sequence of nested $\{\mathbf{C}_i\}_{i=1}^n$, where each $\{\mathbf{C}_i\}$ contains all preceding tokens. These progressively expanding subsets are passed through the model to extract activations at each step. Next, the ensemble classifier $\mathcal{S}_{\text{ensemble}}$ predicts whether the current context $\{\mathbf{C}_i\}_{i=1}^n$ is sufficient.

The activations of all processed chunks are cached to avoid redundant computation. Let $\mathbf{A}_{\text{cache}}^i$ denote cached activations for $\mathbf{C}_i$, containing the activations of all previous chunks by construction. Activations for the current $\mathbf{C}_i$ are computed as:

$$\mathbf{A}(\mathbf{C}_i) = f_{\text{model}}(\mathbf{C}_i \setminus \mathbf{C}_{i-1}, \mathbf{A}_{\text{cache}}^{i-1}),$$

where $f_{\text{model}}$ is the model's forward pass function conditioned on the cached activations $\mathbf{A}_{\text{cache}}$. The iterative process continues until a sufficient context $\mathbf{C}_k$ is identified, as determined by $\mathcal{S}_{\text{ensemble}}$. If no $\mathbf{C}_i$ is deemed sufficient, the entire input context is processed. In either cases, the cached activations will be reused for generation. We discuss potential KV cache optimization in Appendix F.4. The final output is computed as:

$$\mathcal{M}(\mathbf{C}_k) = \mathcal{M}(\mathbf{C}_k \setminus C_{k-1}, \mathbf{A}_{\text{cache}}^{k-1}).$$

**Alternative Sufficiency Detection.** As an alternative to the classifier-based approach, larger LLMs can leverage their own reasoning capabilities through self-prompting. For each cumulative context $\mathbf{C}_i$, we append a meta-prompt asking the model to evaluate whether it has sufficient information to answer query $q$. The prompt can be found in Appendix D.1. The model's binary response determines sufficiency, enabling dynamic cutoff without classifiers. In §3.2, we show that self-prompting becomes increasingly reliable with larger model sizes (14B+ parameters), suggesting that sufficiency detection emerges as a capability with scale.

# 3 Experiments

In this section, we describe our experimental setup in §3.1 and present comprehensive results demonstrating the effectiveness of our method in §3.2.

## 3.1 Experimental Setup

**Datasets.** We use two types of datasets, **single-hop** and **multi-hop**, to assess models' ability to locate the key information across varying context structures and tasks. For single-hop reasoning, where answers are typically found within a single passage requiring minimal context dependency, we use **SQuAD** [19], a widely used dataset with questions based on Wikipedia passages; **Natural Questions** [11], containing questions derived from real-world search queries with answers located in a single but longer passage; and a **Code Understanding** dataset, where we use GPT-4o to synthetically generate multiple single-function code snippets as distractors, and use the original PCSD [26] data to create a QA task dataset requiring to first locate and then understand the relevant code. For multi-hop reasoning, which requires combining information from multiple parts of the context to arrive at the correct answer, we use **HotpotQA** [31], a popular dataset with multi-hop questions requiring reasoning across multiple paragraphs from Wikipedia; **MUSIQUE** [24], a dataset with compositional and nested questions requiring multi-step reasoning across multiple documents; and **Multi-hop Key-Value Retrieval** [36], a widely adopted synthetic dataset for evaluating long-context LLMs that requires exact retrieval of dependent key-value pairs across multiple documents.

**Data Processing.** To evaluate LLMs' long-context capabilities, we extend these naturally short datasets to approximately 40K tokens. Following Liu et al. [16] and Zhang et al. [36], we create long-form versions by combining multiple unique documents within each dataset, conducting experiments on both versions (§3.2). For each data point, we define the ground-truth sufficiency cutoff as the normalized position of the *last* token in the gold information span – the minimal context required for correct answers. Importantly, our evaluation datasets are carefully balanced by design: the gold answer locations follow a uniform distribution (mean $\approx 0.50$, standard deviation 0.25-0.28) across all datasets, ensuring approximately 50% of chunks are classified as "insufficient" and 50% as "sufficient." This balanced distribution prevents bias toward early- or late-context answers and provides a fair assessment of context sufficiency detection. Detailed dataset statistics are provided in Appendix A. We examine the inference time-context length trade-off in our method (§4.2). This definition treats sufficiency as a dataset property, implying a universal cutoff point across models. However, in practice, different models may need varying amounts of context to generate accurate responses, where sufficiency could be model-dependent. We investigate this phenomenon in §4.4. Additionally, when ground truth cutoff points are not available, we show that synthetically LLM generated sufficiency labels are effective proxies in Appendix A.4.

**Models and Baselines.** We evaluate four open-source LLM families ranging from 1B to 70B parameters: LLaMA-3.2-1B [3], Mistral-8B [17], Qwen-2.5-14B [30], and LLaMA-3.3-70B [3]. Note that our proposed method is model-agnostic and can be applied to any Transformer-based LLM. To ensure fair comparison, we follow previous work [10, 18] and evaluate our method against several well-established baselines. For retrieval-based methods (RAG), we include **BM25** [21], which ranks chunks by term frequency, and **SBERT** [20], which uses transformer-based embeddings for semantic relevance. For compression-based methods, we evaluate **LLMLingua** [8], which removes low-entropy tokens; **LongLLMLingua** [10], which applies hierarchical filtering for long contexts; and **LLMLingua2** [18], which learns task-agnostic compression through knowledge distillation. Additionally, we include a **Fine-Tuned Classifier** baseline that learns to predict context sufficiency (Appendix F.3) and **Self-Prompting**, where the LLM assesses sufficiency through prompting (§2.3).

**Evaluation Metrics.** We evaluate two aspects of our dynamic context cutoff method: **sufficiency classification** and **task performance**. For sufficiency classification, we use **F1 Score**, which balances precision and recall, capturing the trade-off between false positives (overestimating sufficiency) and false negatives (underestimating sufficiency). We also report **Recall at 90% Precision (R@90P)**, which measures the percentage of sufficient contexts correctly identified while maintaining a precision of at least 90%. This ensures that the method reliably avoids excessive false positives while achieving high recall. For task performance, we evaluate **Accuracy**, which measures the percentage of correct model outputs against the ground truth after and before context cutoff. We also evaluate **Token Reduction**, which quantifies the proportion of tokens processed relative to the full context. Following previous work [33], we perform model-based evaluation for accuracy calculation by using GPT-4o Mini.[2] More details about the evaluation metrics and prompts can be found in Appendix C and Appendix D.2, respectively.

**Implementation Details.** The proposed dynamic context cutoff method involves three hyperparameters: the classification threshold $\tau$, the number of attention heads used for training, and the number of classifiers in the ensemble. Among these, $\tau$ is the key hyperparameter as it directly impacts the trade-off between efficiency and performance, as discussed in §4.1. The remaining two hyperparameters are determined empirically via a standard hyperparameter sweep on the validation set; details can be found in Appendix F.1. Specifically, we set $k = 5$ for attention heads with the highest F1 scores and train 8 lightweight classifiers for each head, selecting the top 4 with the highest AUC scores to form the ensemble. More details can be found in Appendix F.1. For all methods, including the proposed dynamic context cutoff method, we evaluate using percentage-based chunking with a 10% incremental threshold, meaning each chunk contains 10% more of the full context than the previous one. We explore different chunking strategies in Section 4.2.

## 3.2 Results

**Sufficiency Classification.** Table 1 shows that probing internal attention heads achieves superior sufficiency detection (F1 = 91.1) compared to supervised fine-tuning (79.5) and self-prompting (83.1) in 70B models, demonstrating that latent sufficiency signals are more reliable than surface-level outputs. We also observe an interesting phenomenon: while 1B models struggle with self-prompting (F1 = 52.6), 70B versions achieve much higher performance, suggesting larger models intrinsically develop self-

Table 1: Probing (ours) achieves highest F1 scores compared to supervised fine-tuning (FT) and self-prompting across all models.

| Model | FT | Prompt | Ours |
|---|---|---|---|
| LLaMA3.2-1B | | 52.6 | 88.3 |
| Mistral-8B | 79.5 | 69.7 | 89.8 |
| Qwen2.5-14B | | 78.3 | 87.2 |
| LLaMA3.3-70B | | 83.1 | 91.1 |

assessment capabilities. However, our probing approach maintains consistent high performance across all model sizes, confirming that internal activation provides the most reliable sufficiency detection regardless of model scale.

**Efficiency vs. Performance.** Figure 4 shows the efficiency and performance trade-off between different methods. Unlike static methods (RAG and the Lingua family) that require predefined compression rates or top-$k$ document selection, dynamic methods (FT, self-prompting, and our approach) adaptively determine cutoff points based on content understanding. For our proposed method, we sweep through 4 different $\tau$ values. For 1B models, our method matches RAG and LLMLingua2 in token reduction and achieves comparable accuracy. At 8B, it processes about $1.5\times$ fewer tokens with minimal accuracy drop, outperforming all baselines. For 14B+ models, the method not only reduces tokens up to $1.22\times$ but also improves accuracy. In contrast, RAG degrades sharply with scale. FT underperforms universally, likely due to misaligned sufficiency signals across models. Interestingly, larger models (14B+) exhibit emergent self-awareness via prompting, whereas smaller models (1B-8B) perform poorly in prompting, as instruction-following ability is critical for self-prompting to work effectively. The results suggest that context truncation may also mitigate the "lost-in-the-middle" problem [16, 6], as models focus more on the end of the context, which is likely to contain key information after removal.

---

[2]gpt-4o-mini-2024-07-18

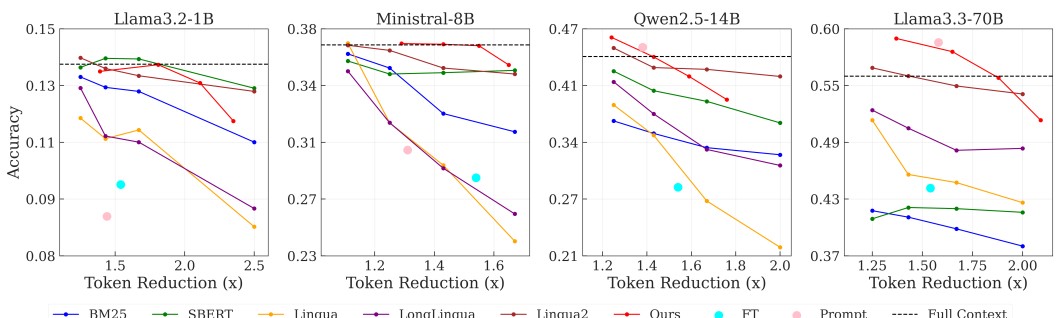

Figure 4: Our method achieves superior efficiency-accuracy trade-offs compared to baselines. RAG degrades with scale, while Lingua2 remains competitive but lags on multihop tasks. Larger models (14B+) exhibit emergent self-awareness on context sufficiency through prompting.

Table 2: Performance comparison across different models on Single-hop and Multi-hop tasks on the *short-form* dataset. Our method achieves a token reduction of $1.33\times$, while outperforming static methods with a targeted compression rate at $1.25\times$.

| Method | LLaMA3.2-1B | | | Ministral-8B | | | Qwen2.5-14B | | | LLaMA3.3-70B | | | Avg. | | |
|---|---|---|---|---|---|---|---|---|---|---|---|---|---|---|---|
| | Multi | Single | Avg | Multi | Single | Avg | Multi | Single | Avg | Multi | Single | Avg | Multi | Single | Total |
| Full Context | 10.4 | 17.9 | 14.2 | **29.6** | 44.8 | 37.2 | 30.4 | 57.6 | 44.0 | 37.1 | 75.0 | 56.1 | 26.6 | 48.7 | 37.9 |
| BM25 | **11.2** | 16.2 | 13.7 | 20.8 | 27.5 | 35.6 | 25.8 | 40.8 | 36.5 | 21.7 | 37.1 | 41.7 | 19.9 | 30.4 | 31.9 |
| SBERT | 10.2 | 17.8 | 14.0 | 19.6 | 37.5 | 35.2 | 26.3 | 51.3 | 42.3 | 22.1 | 41.7 | 40.8 | 19.6 | 37.1 | 33.1 |
| LLMlingua | 6.3 | 18.3 | 12.3 | 22.1 | 41.7 | 31.9 | 24.2 | 52.5 | 38.3 | 35.8 | 74.1 | 55.0 | 22.1 | 46.7 | 34.4 |
| LongLLMlingua | 6.7 | 20.0 | 13.3 | 22.1 | 41.7 | 31.9 | 26.3 | 55.8 | 41.1 | 35.4 | 71.7 | 53.6 | 22.6 | 47.3 | 35.0 |
| LLMlingua2 | 7.9 | **20.8** | **14.4** | 28.3 | 43.3 | 35.8 | 32.1 | 57.9 | 45.0 | 35.8 | 75.0 | 55.4 | 26.1 | 49.3 | 37.7 |
| FT | 6.2 | 13.8 | 10.0 | 21.5 | 34.7 | 28.1 | 22.3 | 35.1 | 28.7 | 35.6 | 52.4 | 44.0 | 21.4 | 34.0 | 27.7 |
| Self-Prompt | 6.4 | 11.4 | 8.9 | 23.8 | 36.2 | 30.0 | **38.2** | 52.0 | 45.1 | **48.3** | 69.9 | 59.1 | 28.9 | 42.6 | 35.8 |
| *Ours* | 10.3 | 17.5 | 13.9 | 28.8 | **45.8** | **37.3** | 33.3 | **59.2** | **46.3** | 43.8 | **75.3** | **59.5** | **29.0** | **49.4** | **39.2** |

Table 3: Performance comparison across different models on Single-hop and Multi-hop tasks on the *long form* dataset. Our method achieves a token reduction of $1.27\times$, while outperforming static methods with a targeted compression rate at $1.25\times$

| Method | LLaMA3.2-1B | | | Ministral-8B | | | Qwen2.5-14B | | | LLaMA3.3-70B | | | Avg. | | |
|---|---|---|---|---|---|---|---|---|---|---|---|---|---|---|---|
| | Multi | Single | Avg | Multi | Single | Avg | Multi | Single | Avg | Multi | Single | Avg | Multi | Single | Total |
| Full Context | 5.0 | 10.4 | 7.7 | 18.3 | **38.8** | 28.5 | 29.9 | 40.0 | 35.0 | 29.3 | 70.8 | 50.0 | 20.6 | **40.0** | 30.3 |
| BM25 | **5.7** | 12.2 | 8.9 | **20.9** | 38.7 | **29.8** | 30.2 | 39.2 | 34.7 | 28.7 | 68.7 | 48.7 | **21.3** | 40.0 | 30.5 |
| SBERT | 5.6 | **12.5** | **9.1** | 20.2 | 37.7 | 29.4 | 30.0 | 38.9 | 34.4 | 27.7 | 68.8 | 48.3 | 21.1 | 39.5 | 30.3 |
| LLMlingua | 3.8 | 12.1 | 7.9 | 17.1 | 35.8 | 26.5 | 27.1 | 41.8 | 34.5 | 23.3 | 65.4 | 44.4 | 17.8 | 38.8 | 28.3 |
| LongLLMlingua | 3.3 | 12.1 | 7.7 | 15.0 | 37.1 | 26.0 | 28.0 | 40.1 | 34.1 | 27.5 | 67.9 | 47.7 | 18.5 | 39.3 | 28.9 |
| LLMlingua2 | 2.7 | 9.6 | 6.2 | 17.1 | 38.3 | 27.7 | 28.8 | 42.9 | 35.8 | 28.2 | 69.2 | 48.7 | 19.2 | 40.0 | 29.6 |
| FT | 2.6 | 8.4 | 5.5 | 14.9 | 31.5 | 23.3 | 21.4 | 33.2 | 27.3 | 21.4 | 47.1 | 34.3 | 15.1 | 30.0 | 22.6 |
| Self-Prompt | 4.2 | 7.3 | 5.7 | 17.4 | 32.6 | 25.0 | **30.5** | **45.8** | **37.6** | 29.0 | 65.4 | 47.2 | 20.3 | 37.8 | 29.0 |
| *Ours* | 5.0 | 9.9 | 7.5 | 19.1 | 37.7 | 28.4 | 29.8 | 43.2 | 36.5 | **30.8** | **70.9** | **50.9** | 21.2 | 39.9 | **30.8** |

**Individual Task Performance.** Table 2 shows that our method maintains consistent performance on both single-hop tasks (49.4% average accuracy) and multi-hop tasks (29%), outperforming the top static baseline, LLMLingua2, by +1.5% in absolute accuracy score. In contrast, RAG methods experience a considerable drop in both settings. For a fair comparison, we report RAG results only for $k = 8$, which corresponds to a compression rate of 0.8 for the Lingua family or a token reduction factor of $1.25\times$. Note that dynamic methods stop naturally and do not target a specific token reduction rate. Our method achieves a $1.33\times$ reduction in tokens, while the FT and Prompt methods achieve reductions of $1.54\times$ and $1.42\times$ on average, respectively.

**Long Context Scenario.** We evaluate our method with longer contexts to assess its scalability. For fair comparison, static methods are evaluated at a fixed $1.25\times$ token reduction. Table 3 shows that our method consistently outperforms baselines, especially in the multi-hop setting. Dynamic methods adaptively determine cutoff points, with FT achieving $1.61\times$, Self-Prompt $1.41\times$, and our method

1.27×, ensuring minimal performance loss. Notably, RAG performs better in long-context settings, particularly for multi-hop reasoning. However, FT remains the weakest method, struggling with generalization. Self-Prompting improves with model size, as larger models better follow instructions for self-assessment. The results confirm that dynamic cutoff outperforms static heuristics. For longer contexts, our method provides an alternative and scalable solution for efficient inference.

# 4 Analysis and Discussion

## 4.1 Classification Threshold

The balance between the model's prediction confidence and the classification threshold $\tau$ is a key factor in our proposed method. In Figure 5, we plot the model's prediction confidence averaged over different numbers of chunks. We observe that the confidence in sufficiency predictions grows steadily as more context is processed, which indicates that useful signals are accumulating over the chunks. Consequently, once the model's confidence exceeds $\tau$, it has likely integrated enough information. Note that stopping too early can cause information loss when critical elements of the context are excluded.

Although the F1 score is a useful measure for detecting context sufficiency, we also report Recall at high Precision to show how well our method identifies truly sufficient contexts while minimizing false positives. In Figure 6, we show results at 90% precision and provide further findings at 95% and 98% precision in the appendix. This metric measures the fraction of actually sufficient contexts that are correctly identified when precision is at least 90%. Such a metric is critical for our task, as a mistaken early cutoff (false positive) can exclude relevant content and degrade the final performance.

## 4.2 Chunking and Inference Time

Chunking determines how efficiently the model processes and evaluates context sufficiency. Table 4 compares different chunking strategies for Qwen-2.5-14B. Percentage-based chunking performs consistently well, with 10% chunking offering the best trade-off between accuracy and efficiency. While sentence-level chunking achieves the highest classification performance, it is impractical due to the increased overhead of frequent sufficiency checks. Since these checks require processing chunks sequentially, smaller chunks lead to higher latency, as each additional step incurs computational overhead before reaching a decision even with caching.

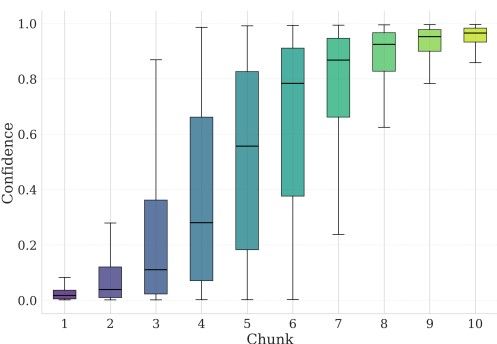

Figure 5: Confidence progression across context chunks. Model's prediction confidence increases monotonically with more context.

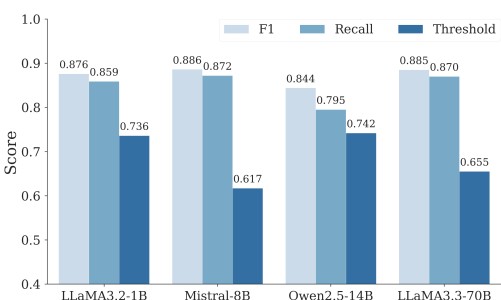

Figure 6: F1 score and Recall at 90% precision for sufficiency detection. Our approach reliably identifies when enough context is present while minimizing false positives. More results can be found in Appendix G.

Table 4: Sentence-level chunking achieves the highest performance but is computationally expensive. 10% chunking offers the best balance between accuracy and efficiency.

| Metric | Sent. | 1% | 5% | 10% | 20% |
|---|---|---|---|---|---|
| F1-Score | 96.8 | 87.2 | 87.0 | 88.3 | 88.3 |
| R@90P | 95.4 | 90.9 | 78.4 | 85.9 | 85.8 |
| Acc. | 14.5 | 13.7 | 12.8 | 13.9 | 13.7 |

Therefore, 10% chunking is chosen to best balance granularity and efficiency. Figure 7 shows inference time between our method (10% chunking) and full-context processing. For short contexts (1K tokens), directly processing the full context is faster; however, beyond 2K tokens, our method provides significant inference time savings when fewer than six chunks (60% of the full context) are processed. This demonstrates that our approach scales efficiently, offering increasing benefits for longer inputs.

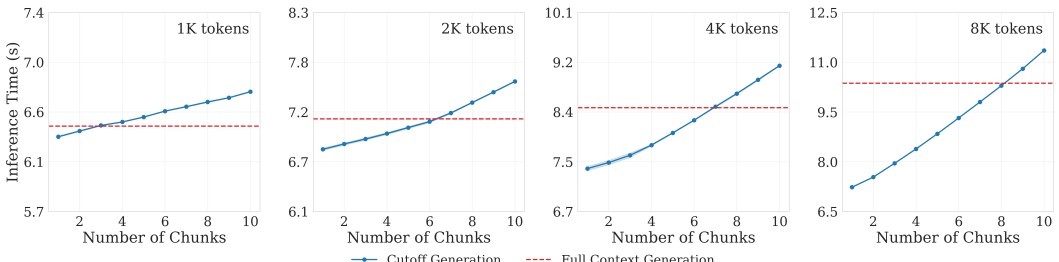

Figure 7: For short contexts (1K tokens), full-context processing is faster. However, beyond 2K tokens, our method becomes more efficient, achieving faster inference when fewer than six chunks (60% of the full context) are processed.

## 4.3 Wall-Clock Time vs. Accuracy

Beyond token reduction, we compare wall-clock time in Table 5, using the same configuration from Section 4.2. All experiments were run on the same hardware configurations as detailed in Appendix F.2. Our method achieves faster inference time than full context processing while also improving accuracy from 35.0% to 36.5%. LLMLingua2 is the fastest overall at 6.97s with comparable accuracy of 35.8%. Self-Prompting, while achieving the highest accuracy (37.6%), is the slowest among all methods. RAG methods (BM25 and SBERT) and other Lingua variants offer some speedup over full context but generally at the cost of accuracy. The FT method achieves faster inference than full context.

Table 5: Comparison of average wall-clock inference time (seconds per sample) and average accuracy across various methods.

| Method | Time (s) | Acc. (%) |
|---|---|---|
| Full | 9.02 | 35.0 |
| BM25 | 8.68 | 34.7 |
| SBERT | 8.93 | 34.4 |
| LLMLingua | 7.35 | 34.5 |
| LongLLMLingua | 8.47 | 34.1 |
| LLMLingua2 | 6.97 | 35.8 |
| FT | 8.01 | 27.3 |
| Self-Prompt | 10.5 | 37.6 |
| Ours | 8.13 | 36.5 |

However, it results in a significant drop in accuracy. Overall, our method offers a balanced trade-off, reducing latency without external heuristics while preserving answer quality.

## 4.4 Universal vs. Model-Specific Cutoffs

From a human perspective, each task has a "gold" location in the context where the final relevant information resides—once an answer is directly obtained, any further context is redundant. In such cases, a universal stopping point may be plausible. However, from a model perspective, defining a single optimal cutoff is challenging and ambiguous. For example, in in-context learning (ICL), models observe demonstration examples without a clear threshold for sufficiency. Smaller models may require more examples to generalize, while larger models may reach high confidence with fewer. This suggests a model-specific cutoff, where each model determines its own stopping threshold rather than adhering to a universal standard. This is particularly relevant in real-world applications, where different LLMs and tasks have varying context requirements. We provide preliminary exploration of tasks without explicit answer locations in Appendix E, using ICL as a representative case.

## 4.5 Beyond Factoid QA

Our work focuses on tasks where the information needed to answer a query is localized within specific parts of the context. While this represents a substantial portion of real-world applications (e.g., question answering, information retrieval, fact verification), we acknowledge that not all tasks benefit from early stopping. Tasks requiring holistic understanding of the entire context, such as summarization or passage rewriting, may not be suitable candidates for dynamic cutoff. However, a key advantage of our method is its ability to handle both scenarios naturally. Unlike compression methods that reduce context regardless of task requirements, our sufficiency classifier can process the full context when necessary – when all information is crucial, the classifier would not trigger early stopping, effectively using the entire input. We also demonstrate in Appendix A.4 that synthetically generated sufficiency labels (via GPT-4o) achieve competitive performance (F1: 84.6-87.0 vs 88.3-

89.8 for original labels), enabling extension beyond factoid QA. Additionally, for larger models (14B+), our self-prompting approach eliminates dependence on labeled data entirely, suggesting potential for broader task coverage.

### 4.6 Limitations and Future Work

While our sufficiency classifier demonstrates promising generalization through synthetic labels and self-prompting, its applicability to all task types (e.g., creative writing, open-ended dialogue) remains an open question. Future work could investigate classifier performance across broader task spectrums and develop adaptive threshold selection mechanisms that automatically adjust $\tau$ based on model characteristics and task requirements, rather than relying on validation-based hyperparameter tuning.

## 5 Conclusion

We introduce dynamic context cutoff, a method that enables LLMs to process only the minimal necessary context by detecting context sufficiency signals using the model's internal representations. This approach reduces token processing by $1.33\times$ on average while improving accuracy by $3.4\%$, outperforming static methods like RAG and compression-based heuristics. We find that larger models develop emergent self-assessment capabilities, allowing them to detect sufficiency through self-prompting. By enabling models to terminate processing dynamically, our method enhances efficiency and scalability for LLM inference, paving the way for more intelligent context processing.

## Acknowledgements

We thank Shuyan Zhou, Sanxing Chen, Raghuveer Thirukovalluru, Saloni Potdar, and Dong Lin for thoughtful initial discussions, and all other members of the DukeNLP lab for their valuable feedback. Roy Xie is supported by the Apple Scholars in AI/ML PhD fellowship and NSF Graduate Research Fellowship. This work was also supported by the Learning Engineering Virtual Institute, funded by leading education philanthropists and organizations through Grant G-23-2137070 to the University of Florida and its partner institutions. The opinions expressed are those of the authors and do not represent the views of the universities, institutions, or those of the philanthropists and organizations.

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

# Appendix

# A  Dataset

## A.1  Statistics

We define gold location as the gold information span's end position divided by the total number of tokens in the input, which is uniformly distributed. We provide statistics for the Long datasets as shown below in Table 6 and Short-form dataset in Table 7. Each dataset contains 600 data points, and the train-validation-test split is 80%, 10%, and 10%, respectively.

Table 6: The datasets are grouped based on reasoning type. Code Understanding (Code) is distinct as it involves synthetic code understanding, whereas SQuAD and Natural Questions focus on retrieving answers from a single passage. Multi-hop Key-Value Retrieval (KV) is separate as a synthetic multi-hop key-value retrieval task, while HotpotQA and MUSIQUE involve natural multi-hop reasoning across multiple passages. Statistics include Token Count and Gold Location.

| Statistic | Single-hop | | Multi-hop | |
| --- | --- | --- | --- | --- |
| | **Code** | **SQuAD & Natural Questions** | **KV** | **HotpotQA & MUSIQUE** |
| *Token Count* | | | | |
| Mean | 22,758.96 | 22,653.68 | 22,494.19 | 24,970.78 |
| Median | 22,739.50 | 22,584.00 | 22,619.00 | 25,168.50 |
| Std Dev | 7,299.95 | 6,758.97 | 7,290.87 | 7,159.93 |
| Max | 35,465 | 36,888 | 35,426 | 40,622 |
| Min | 10,107 | 9,955 | 10,023 | 10,430 |
| *Gold Location* | | | | |
| Mean | 0.50 | 0.50 | 0.51 | 0.49 |
| Median | 0.50 | 0.50 | 0.49 | 0.49 |
| Std Dev | 0.28 | 0.28 | 0.25 | 0.26 |
| Max | 0.99 | 0.99 | 0.96 | 0.96 |
| Min | 0.01 | 0.01 | 0.04 | 0.02 |

Table 7: Categorized by reasoning type, with single-hop tasks (Code Understanding, SQuAD, and Natural Questions) involving direct retrieval from a passage, and multi-hop tasks (Multi-hop Key-Value Retrieval, HotpotQA, and MUSIQUE) requiring inference across multiple segments. Statistics include both token count and gold location.

| Statistic | Single-hop | | Multi-hop | |
| --- | --- | --- | --- | --- |
| | **Code** | **SQuAD & Natural Questions** | **KV** | **HotpotQA & MUSIQUE** |
| *Token Count* | | | | |
| Mean | 2951.23 | 1678.98 | 2723.65 | 1676.15 |
| Median | 2910.50 | 694.50 | 2704.00 | 1735.00 |
| Std Dev | 562.14 | 1269.42 | 450.02 | 646.41 |
| Max | 4990 | 4993 | 4616 | 2815 |
| Min | 1489 | 600 | 1572 | 578 |
| *Gold Location* | | | | |
| Mean | 0.51 | 0.48 | 0.56 | 0.53 |
| Median | 0.52 | 0.49 | 0.55 | 0.50 |
| Std Dev | 0.23 | 0.28 | 0.20 | 0.28 |
| Max | 0.98 | 0.99 | 0.91 | 0.99 |
| Min | 0.02 | 0.02 | 0.21 | 0.02 |

## A.2  Dataset Balance

Our information sufficiency evaluation dataset is carefully balanced by design. As described in Appendix A, the gold location in our context is sampled from a uniform distribution, placing the required information approximately at the middle of the context. This creates a balanced evaluation where approximately 50% of context chunks are classified as "insufficient" (before the gold location)

and 50% as "sufficient" (after and including the gold location). This balanced distribution ensures that our evaluation is not biased towards either early or late stopping decisions, providing a fair assessment of the method's ability to detect context sufficiency.

## A.3 Sufficiency Label Collection Process

To train our sufficiency detection classifiers, we create datapoints by labeling context chunks as either sufficient or insufficient for answering the given question. Here we describe our methodology for generating these sufficiency labels.

**Label Generation Process.** We generate sufficiency labels by first splitting the context into non-overlapping chunks according to our chunking strategy (e.g., 10% of total tokens per chunk). Using the ground truth answer location(s), we identify the answer-containing chunk(s) in the document. We then label all chunks that appear before the answer-containing chunk as insufficient (0), while marking the answer-containing chunk itself and all subsequent chunks as sufficient (1). This labeling approach is based on the intuition that a question becomes answerable *if and only if* all necessary information chunks are present in the context. Note that the labeling process varies slightly for different question types:

- **Single-hop Questions:** These typically require information from a single passage or section within the document. Depending on the chunking strategy, there is usually only one answer-containing chunk. All chunks before this are labeled as insufficient, while this chunk and all subsequent chunks are labeled as sufficient.

- **Multi-hop Questions:** These questions require integrating information from multiple parts of the document. There may be multiple answer-containing chunks (e.g., different pieces of information needed from different sections). In these cases, only the last answer-containing chunk and all subsequent chunks are labeled as sufficient, as all required information is only available after that point.

## A.4 Synthetic Sufficiency Labels

Most existing QA datasets (including all six datasets used in our paper) are constructed with known answer locations, making it straightforward to generate sufficiency labels as described above. However, this approach may not be directly applicable to scenarios where answer locations are not explicitly provided. To address this limitation, we investigated whether large language models could generate synthetic sufficiency labels that perform comparably to those derived from human-annotated ground truth locations. We conducted experiments comparing classifiers trained with two types of labels:

- **Original Labels:** Generated using the ground truth answer locations as described above.

- **Synthetic Labels:** Generated using GPT-4o to predict answer locations within the documents.

For the synthetic label generation, we prompt GPT-4o to identify the minimal set of context chunks required to answer each question completely. We then use these predictions to label chunks as sufficient or insufficient following the same methodology used for original labels. Table 8 shows the performance comparison between classifiers trained with synthetic versus original labels. The evaluation pipeline for both remained identical, relying on the same ground-truth labels for testing.

The results show that while there is a modest performance gap, classifiers trained with synthetic labels still achieve strong performance that is competitive with those trained on original labels. This indicates that our approach can be effectively extended to scenarios where explicit answer locations are not available, by leveraging LLMs to generate reasonably accurate sufficiency labels.

Table 8: Performance comparison between classifiers trained with synthetic (GPT-4o generated) versus original (human-annotated) sufficiency labels across different model sizes.

| Task Type | Synthetic | | Original | |
|---|---|---|---|---|
| | 1B | 8B | 1B | 8B |
| Single-hop | 82.1 | 84.4 | 85.7 | 89.3 |
| Multi-hop | 87.1 | 89.6 | 90.9 | 90.3 |
| Overall F1 | 84.6 | 87.0 | 88.3 | 89.8 |
| P90 | 79.3 | 82.7 | 85.9 | 90.1 |

## B  Additional Probing Details

### B.1  Activation Head Selection

For efficient context cutoff, our method does not use activations from all layers of the model, but rather selectively identifies the most informative attention heads in specific layers through probing. The activation selection process works as follows:

- We initially probe all attention heads across all layers of the model to identify which ones encode the strongest sufficiency signals.

- As shown in Figure 3 for LLaMA3.2-1B and Figure 8 for Qwen2.5-14B, we discovered that a subset of heads, primarily from middle layers, exhibit significantly higher predictive performance for context sufficiency. This aligns with findings in other interpretability work that middle layers often capture higher-level semantic information.

- After identifying these predictive heads, we select only the top-$k$ heads with the highest F1 scores on the validation set ($k = 5$ in our implementation) to train our ensemble classifier.

As demonstrated in Table 9, we found that using just the top 5 attention heads yields the best performance, with minimal gains or even decreased performance when more heads are included. This confirms our hypothesis that context sufficiency signals are concentrated in specific architectural components rather than distributed throughout the entire model.

The specific layers used can vary across model architectures - we don't restrict our approach to predetermined layers, but rather let the probing results guide which heads (and consequently which layers) provide the most reliable sufficiency signals. This approach ensures our classifier focuses only on the most informative components of the model's internal representations while keeping computational overhead minimal.

### B.2  Additional Probing Results

Figure 8 illustrates the probing results for the Qwen2.5 14B model, revealing that, similar to LLaMA models, the highest F1 scores are concentrated in the middle layers. However, the distribution of these high-performing heads differs between the two model families. While both models exhibit darker regions indicating stronger sufficiency signals in their intermediate layers, LLaMA3.2-1B shows a more dispersed pattern of high F1 scores across various heads within these layers. This suggests that although both LLaMA and Qwen models tend to encode context sufficiency signals primarily in their middle layers, the specific attention heads responsible and their activation patterns vary between architectures.

### B.3  Left-to-Right Context Processing

Our choice of left-to-right processing is motivated by two main factors. First, Transformer models are typically trained on left-to-right sequences, making this order naturally compatible with their internal representations. This avoids the need for significant architectural changes or retraining. Second, it enables efficient use of the key-value (KV) cache while preserving semantic consistency. As described in §2.3, left-to-right processing allows us to reuse cached activations from previous

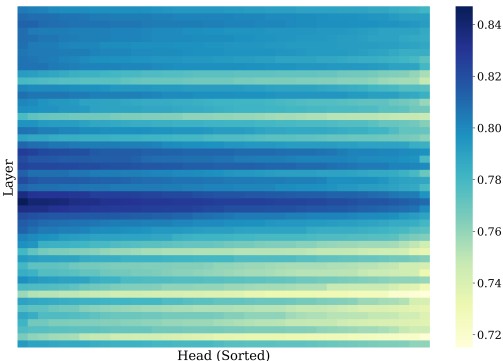

Figure 8: Probing results for the Qwen2.5 14B model. The heatmap shows the average F1 score for each head across all layers, which is different from LLaMA models.

chunks, maintaining contextual coherence across the sequence. As shown in Figure 5, the model's confidence in sufficiency predictions increases steadily as more context is processed, suggesting that meaningful information accumulates effectively under left-to-right processing.

Although alternative processing orders (e.g., reversed or random) or methods (e.g., RAG, which selects arbitrary subsets of context) are possible, our approach preserves semantic continuity between chunks. We leave the investigation of these alternatives for future work.

Note that when processing a chunk, we retain the KV cache from all preceding chunks. This means there is no computational difference between processing the context chunk-by-chunk from left to right and processing the entire context in a single pass. We do not alter the computation over the context; we simply segment it into chunks to determine when to stop. Each token receives exactly the same context as it would without chunking, due to reuse of the KV cache.

## C   Evaluation Metrics

Our evaluation framework employs two categories of metrics to comprehensively assess different aspects of our method:

- **Information Sufficiency Classification metrics**:
  - F1 Score: Measures the overall balance between precision and recall in detecting sufficient context. This metric is particularly important as it penalizes both false positives (stopping too early) and false negatives (processing unnecessary context). A high F1 score indicates that our method can reliably identify when enough information has been processed while avoiding premature cutoffs.
  - Recall at 90% Precision (R@90P): Ensures high confidence in sufficiency predictions while maintaining good coverage. This metric is crucial for our task as it measures how many truly sufficient contexts we can identify while keeping false positives (incorrect early cutoffs) below 10%. This conservative approach helps prevent information loss while still achieving efficiency gains.

- **QA Task Performance metrics**:
  - Accuracy: Measures answer correctness before and after context cutoff. This metric is calculated as the percentage of questions answered correctly by comparing model outputs with ground truth answers. We use GPT-4o Mini as an automated judge to evaluate answer correctness, following established practices in QA evaluation [33]. This approach is more reliable than exact string matching, especially for long-form answers where semantic equivalence is more important than lexical matching.
  - Token Reduction: Quantifies the proportion of tokens processed relative to full context. This metric directly measures computational efficiency gains, calculated as the ratio between the number of tokens processed with our method versus processing the full context. A higher token reduction indicates greater computational savings while improving performance.

# D Prompts

## D.1 Self-Sufficiency Prompt

> **Self-Sufficiency Prompt**
>
> Given the following context and question, determine if the context contains enough information needed to answer the question.
>
> [QUESTION]: {question}
>
> [CONTEXT]: {context}
>
> Your response should strictly ONLY consist of '[[YES]]' if context is enough, or '[[NO]]' if context is not enough. Omit any other output.
>
> Your response:

## D.2 Evaluation Prompt

> **Evaluation Prompt**
>
> You are an expert model evaluator specializing in natural language understanding. Your task is to determine if a model's answer is correct by comparing it with the provided gold answers, accounting for valid paraphrasing and alternate expressions of the same answers.
>
> [QUESTION] {question} [/QUESTION]
>
> [GOLD_ANSWERS] {correct_answers} [/GOLD_ANSWERS]
>
> [MODEL_ANSWER] {model_answer} [/MODEL_ANSWER]
>
> Evaluation criteria: - Answer must convey the same core meaning as gold answers - Partial matches should be marked incorrect - Additional correct information beyond gold answers is acceptable - Empty or off-topic responses are incorrect
>
> Your response should strictly ONLY consist of '[[YES]]' if model answers question correctly, or '[[NO]]' if model answers question incorrectly. Omit any other output.
> Your response:

## D.3 Answer Generation Prompt

> **Answer Generation Prompt**
>
> Please provide a response to the query based only on the given context:
>
> [QUESTION]: {question}
>
> [CONTEXT]: {context}
>
> Your response:

# E    Model-Specific Cutoffs

While our work focuses on factoid queries where evidence is localized (data with ground truth sufficient information label), our approach can be extended to rationale queries. We believe that the model's internal representations still encode when it has gathered sufficient information to form a coherent response, even if that information is distributed across the document. Sufficiency is ultimately a property of the model's understanding, not just the dataset structure. In this section, we discuss some preliminary findings and ideas under an in-context-learning setting, which does not have explicit answers located in the context. Furthermore, our experiments show synthetically generated sufficiency labels (via GPT-4o) are effective proxies when explicit answer locations are unavailable Appendix A.4. The modest performance gap indicates potential for rationale queries.

## E.1    Case Study: In-Context Learning

To explore the challenges of defining model-specific cutoffs, we utilized the TREC dataset [5] for In-Context Learning (ICL) task. TREC comprises a series of questions categorized into six distinct types, Abbreviation, Entity, Description and abstract concept, Human being, Location, and Numeric value; these six types are labled from zero to five respectively. Each question type serves as a category label, and the dataset is structured to provide multiple examples per category without revealing these labels to the models. This setup requires the model to generalize from demonstrated examples to accurately classify unseen queries. For our experiments, we employed two models of differing scales: a 8-billion-parameter (8B) Mistral model and a 14-billion-parameter (14B) Qwen2.5 model. These models were selected to illustrate the variance in context processing capabilities across different model sizes, providing insights into how each handles the accumulation of context in an ICL setting.

## E.2    Analysis

Figure 9 presents the probability of outputting labels as the models process sequential examples from the TREC ICL dataset. The 8B model exhibits a gradual increase in confidence, requiring nearly all available examples to achieve its highest accuracy. In contrast, the 14B model reaches peak confidence after processing only a subset of the examples, demonstrating a more rapid understanding of the underlying category structure. This discrepancy highlights that larger models can infer task requirements more efficiently, suggesting that a universal cutoff—applicable to all model sizes—would be suboptimal. The figure also reveals instances where the 8B model remains uncertain despite processing additional examples, whereas the 14B model consistently converges on the correct label with fewer demonstrations. These observations underscore the necessity for model-specific thresholds that account for each model's unique capacity to assimilate and generalize from context.

## E.3    Implications

The variability in cutoff points between the 8B and 14B models in the ICL setting indicates that a one-size-fits-all approach to context cutoff is inadequate for more nuanced tasks. Methods, such as halting after a fixed number of examples or relying solely on confidence thresholds, may lead to inconsistent performance across different model architectures. For instance, early examples in the ICL dataset can sometimes mislead smaller models, causing them to misclassify subsequent queries. Addressing this requires developing adaptive cutoff mechanisms that dynamically adjust based on the model's internal state and the specific characteristics of the task. Future research should focus on designing algorithms that can learn these individualized thresholds, potentially leveraging additional signals from the model's activations or exploring hybrid approaches that combine universal and model-specific criteria. Furthermore, applying such techniques to datasets where the gold information is not easily identifiable will be crucial for validating the robustness and generalizability of model-specific cutoff strategies.

# F    Implementation Details

## F.1    Ensemble Classifier

For the ensemble classifier, the folds are constructed from the training split during cross-validation. The validation split is held out for the evaluation after the classifier is built. Tables 9 shows the perfor-

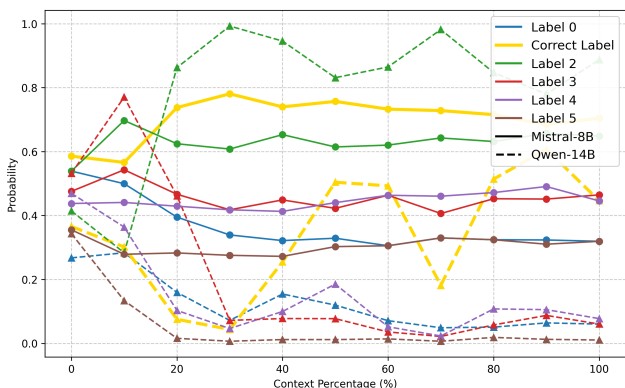

Figure 9: Confidence progression in TREC ICL task: The 8B model requires nearly all examples to achieve its highest confidence, whereas the 14B model attains peak confidence after processing fewer examples. This illustrates the need for model-specific cutoff thresholds.

mance comparison in different number of attention heads and different classifiers used in ensemble. For attention heads, we found that using only the top 5 selected heads yield best performance, and use the top 4 out of 7 classifiers is the best configuration.

Table 9: Performance comparison across head selections and number of classifiers for our method.

| | Head Numbers | | | Classifier Numbers | | |
|---|---|---|---|---|---|---|
| **Metrics** | **5** | **10** | **20** | **2** | **4** | **6** |
| F1-Score | 88.3 | 87.3 | 87.9 | 87.4 | 88.3 | 87.3 |
| R@90P | 85.9 | 78.0 | 78.0 | 77.6 | 85.9 | 78.0 |
| Acc. | 13.9 | 13.0 | 12.8 | 12.7 | 13.9 | 12.9 |

### F.2 Memory Requirements and Computational Requirements

Our ensemble classifier consists of small tree-based and linear models with extremely minimal memory footprints, typically in the range of a few megabytes per model. The full ensemble model consists of 8 linear/tree-based classifiers, from which we select the top 4 with the highest validation F1 scores as our final ensemble. The total memory requirement for our complete ensemble is less than 15MB, which is negligible compared to the multi-gigabyte memory requirements of the LLMs themselves (often 6-140GB depending on model size).

During our experiments, we ran these classifiers on GPUs alongside the LLMs for convenience and faster iteration. We were able to run all experiments (including with 70B models) on just 2-4 A5000/A6000 GPUs (as detailed in Table 10), as the classifier's memory requirements are negligible in the overall GPU memory budget. For deployment scenarios where GPU memory efficiency is particularly important, offloading the classifier to CPU while keeping only the LLM on GPU is a viable option. This approach incurs minimal latency overhead since the classifier's computation is lightweight compared to the LLM's forward pass. We leave the detailed analysis of the memory and latency trade-off for future work.

Table 10: GPU configurations used for different models in our experiments.

| Model | GPUs Used |
|---|---|
| LLaMA 3.2-1B | 2 × Nvidia A5000 |
| Mistral 8B | 4 × Nvidia A5000 |
| Qwen 2.5-14B | 4 × Nvidia A5000 |
| LLaMA 3.3-70B | 4 × Nvidia A6000 |

### F.3 Fine-Tuned Classifier (FT)

We fine-tune meta-llama/Llama-3.2-1B to predict the context cutoff point in long-context inputs, formulating this as a binary classification task. The model is trained on the Short-form dataset specified in Appendix A. We optimize using the AdamW optimizer with a learning rate of 8.0e-05 and a batch size of 32, employing a cosine learning rate schedule with linear warmup. The fine-tuned model achieves a development set accuracy of 0.8346, demonstrating strong predictive capability. We chose meta-llama/Llama-3.2-1B due to its efficiency in capturing long-range dependencies while maintaining manageable computational costs. Additionally, framing the task as binary classification simplifies optimization and enables robust generalization across diverse long-context scenarios. We include meta-llama/Llama-3.2-3B results and the performance of training on long dataset in Table 11 for reference. All models are fine-tuned for one epoch.

Table 11: Performance of fine-tuned classifiers tuned on different datasets.

| Base Model | Trained & Evaluated on | Test Accuracy |
|---|---|---|
| Llama3.2-1b | Short Dataset | 0.8346 |
| Llama3.2-1b | Long Dataset | 0.7515 |
| Llama3.2-3b | Short Dataset | 0.8413 |
| Llama3.2-3b | Long Dataset | 0.7456 |

### F.4 Potential Combination with KV Cache Optimization

Recent work has explored KV cache optimization techniques to improve inference efficiency. As also discussed in §1, while KV cache optimization focuses on reducing or evicting less important KV cache entries to reduce memory usage for decoding speedup, our method reduces initial text processing at the input level (like LLMLingua). This means these approaches are complementary and can be potentially combined - our method reduces input size, and KV cache optimization could further improve decoding speed. While combining both methods could lead to additional efficiency gains, it is beyond the scope of this work. We consider this an interesting direction for future research.

## G  Recall at High Precision

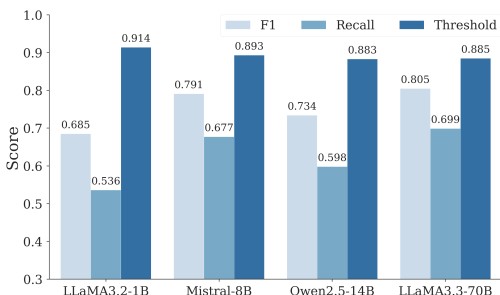

Figure 10: F1 score and Recall at 95% precision (R@95P) for sufficiency detection.

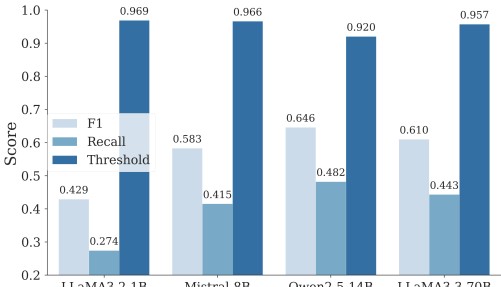

Figure 11: F1 score and Recall at 98% precision (R@98P) for sufficiency detection.

