# OpenReview forum: "Knowing When to Stop: Efficient Context Processing via Latent Sufficiency Signals"
_NeurIPS.cc/2025/Conference — NeurIPS 2025 poster_

### Official Review · Reviewer_2h3k · 2025-06-11

**Clarity:** 4
**Significance:** 2
**Originality:** 2
**Rating:** 4
**Confidence:** 4

**Summary:**

This paper introduces dynamic context cutoff, a novel human-inspired method for large language models (LLMs) that allows them to self-terminate processing once sufficient task-relevant information is acquired. However, the authors claim that *this approach challenges traditional reliance on external compression heuristics, instead leveraging the model's internal understanding to dictate processing needs*, being worth considering, as the method proposed by the authors is a classifier-based approach.

**Questions:**

Could you explain whether the selection of these hyperparameters is sensitive? Could you also clarify if obtaining the classifiers is straightforward? Additionally, can the same classifier be used for judgment across different model families?

**Ethical Concerns:**

["NO or VERY MINOR ethics concerns only"]

**Final Justification:**

Most of the concerns have been eliminated.

**Limitations:**

yes

**Quality:**

3

**Strengths And Weaknesses:**

**strengths**

- The writing of this paper is very clear, and the introduction is also quite detailed.
- The experiments in this paper are abundant, especially the paper has conducted validations on many datasets.

**weaknesses**

- The method proposed in this paper is classifier-based and requires a series of classifiers. the technical innovation of this approach is not particularly high.  In scenarios where the context length is dynamically determined, a substantial number of researchers have already conducted studies and published relevant reports on this topic. Some papers published before the submission deadline are listed below. In contrast, using a classifier for context processing is more straightforward and less flexible.

- since there are three hyperparameters in this paper, the persuasiveness of the experimental results' improvement becomes weaker, and determining how to set these hyperparameters will be an issue.

[1] Wu, Wei, et al. "Tokenselect: Efficient long-context inference and length extrapolation for llms via dynamic token-level kv cache selection." arXiv preprint arXiv:2411.02886 (2024).

[2] Li, Yucheng, et al. "Compressing context to enhance inference efficiency of large language models." arXiv preprint arXiv:2310.06201 (2023).

---

> ### Author Rebuttal · Authors · 2025-07-30
>
> Dear Reviewer 2h3k,
>
> We sincerely thank you for your constructive feedback and recognition of our paper's strengths, including clear writing and comprehensive experiments and evaluations. We appreciate you taking the time to review our paper and would like to address your concerns as follows:
>
> ---
>
> > **Technical Innovation and Core Contributions**
>
> Thank you for this important question! We would like to clarify that there are two orthogonal approaches to efficiency: (1) methods that approximate transformer computations during inference (like TokenSelect, which performs sparse attention by dropping low-attention tokens), and (2) methods that reduce input context length. These approaches are complementary, as any context compression method can benefit from computational approximations like KV optimization.
>
> Our work falls in the *second* category, focusing on context reduction rather than computational approximation. Within this space, we make two novel contributions that distinguish our approach from existing context compression methods:
>
> *First*, we are the first to leverage internal LLM representations to decide when to stop processing, discovering that certain attention heads naturally encode “sufficiency signals” that indicate when enough context has been seen. This is a previously unexplored aspect of transformer interpretability.
>
> *Second*, rather than treating context compression as a separate preprocessing module (as in prior work [1]), we make it an inherent part of how the model encodes text during inference. This integration enables dynamic, model-driven decisions rather than fixed compression rules.
>
> We compare against all relevant baselines within the context compression category. Regarding the Li et al. (2023) reference, we note this was the proof-of-concept version later developed into LLMLingua, and we do compare against the improved LLMLingua methods in our experiments.
>
> Due to page limits, our discussion of the complementary nature of these two categories was moved to Appendix H.4 (Potential Combination with KV Cache Optimization). In the revision, we will expand the discussion in the main text and include both mentioned relevant work in the related work section to better position our contributions. Thank you again for raising this excellent question!
>
> ---
>
> > **Hyperparameter Sensitivity and Selection**
>
> Thank you for this insightful question! Among our three hyperparameters, only the classification threshold τ impacts the efficiency-accuracy trade-off, while the other two (number of heads and ensemble size) are determined through standard hyperparameter search. Importantly, *we use a single global threshold across all datasets (Tables 2 and 3)*, as our method does not require dataset-specific tuning. As shown in Figure 1, different threshold values create predictable efficiency-accuracy trade-offs, but the key finding is that a single global setting works effectively across all experimental conditions. This addresses the core concern about sensitivity - we don't need separate thresholds for each dataset type.
>
> Practically, the threshold τ can be set once based on desired efficiency-accuracy preferences and deployed universally. We also provide detailed sensitivity analysis on number of heads and ensemble size in Appendix B.1 and H.1, showing stable performance across reasonable ranges. Additionally, for larger models, our self-prompting approach eliminates dependence on trained classifiers entirely, which highlights the potential extension of our method with a training-free alternative which does not require hyperparameter tuning.
>
> ---
>
> > **Classifier Complexity and Obtainment**
>
> You raise an important question regarding the ease of obtaining classifiers and their applicability across model families. We clarify that training classifiers is straightforward, requiring only attention head activations and binary sufficiency labels. The training process follows a simple procedure: probe all attention heads, select top-performing heads, and train the classifiers. This one-time setup enables deployment across *all tasks* for that model. The classifiers are lightweight, using linear or tree-based architectures with minimal memory footprint (Appendix H.2). Binary sufficiency labels can be obtained from existing datasets when available (Appendix A.3), or generated using language model judges through prompting (Appendix A.4).
>
> In terms of cross-model applicability, classifiers cannot be directly transferred across model families due to architectural differences in attention mechanisms and internal representations. This model-specific requirement aligns with previous interpretability work [2] and reflects a core research contribution: demonstrating that internal model representations encode sufficiency signals. While extracting these signals requires model-specific classifiers, our probing methodology itself is universal. Once trained, classifiers show consistent performance across tasks within the same model. This model-specific nature also reflects realistic differences in how various architectures process context, as different models exhibit different context utilization patterns (L220-222).
>
> ---
>
> [1] Li, Yucheng, et al. "Compressing context to enhance inference efficiency of large language models." arXiv preprint arXiv:2310.06201 (2023).
>
> [2] Inference-Time Intervention: Eliciting Truthful Answers from a Language Model
>
> ---
>
> Once again, thank you for the thoughtful review and the opportunity to clarify these important aspects of our work. Your questions have helped us identify areas where we can strengthen our presentation and better highlight the novel contributions of our approach. Thank you for your valuable feedback and consideration of our work!
>
> Best regards,
>
> Authors

---

> > ### Comment · Reviewer_2h3k · 2025-08-01
> > **Thank you for your response**
> >
> > Thank you for your reply. I agree with most of your claims, but I'm still confused about "one-time setup". Since dynamic context length seems like a task highly relevant to deployment scenarios, can the Classifiers obtained through such a setup really guarantee their general applicability across all tasks?

---

> > > ### Author Response · Authors · 2025-08-01
> > > **Response to Reviewer 2h3k**
> > >
> > > Dear Reviewer 2h3k,
> > >
> > > Thank you for this insightful question! You are absolutely right that the "one-time setup" phrasing needs clarification. To be precise: we train *one classifier per model* that works across our experimental tasks (multi/single hop QA, retrieval, reading comprehension). We selected these tasks as they represent a reasonable range of QA scenarios with varying complexity and context requirements, providing a representative evaluation of dynamic context cutoff capabilities.
> > >
> > > We acknowledge that this work serves as an initial attempt at dynamic context cutoff rather than aiming for generalization to all possible tasks. However, you raise an important point about broader deployment scenarios. For applications requiring coverage beyond our experimental scope, our self-prompting approach offers a promising training-free alternative that can generalize to new tasks without classifier dependence, which we believe is an interesting avenue for future work.
> > >
> > > Thank you once again for your thoughtful engagement with our work and constructive feedback!
> > >
> > > Best regards,
> > >
> > > Authors

---

> > > > ### Comment · Reviewer_2h3k · 2025-08-01
> > > >
> > > > Thanks for your response/
> > > >
> > > > Now I understand. So this is indeed a limitation that cannot be overlooked.

---

> > > > > ### Author Response · Authors · 2025-08-02
> > > > > **Response to Reviewer 2h3k**
> > > > >
> > > > > Dear Reviewer 2h3k,
> > > > >
> > > > > Thank you for your response. We appreciate your engagement with our work and would like to address the concern about generalization across tasks.
> > > > >
> > > > > Our classifier *already* demonstrates good cross-task generalization, as evidenced by consistent performance using *one classifier with one global threshold* across all evaluated tasks (Tables 2 and 3). We would also like to clarify the following important points:
> > > > >
> > > > > **For tasks without explicit answer locations:** We provide several practical solutions in our paper. In Appendix A.4, we conduct experiments using classifiers trained on synthetic versus original labels. Our results show that LLM-generated synthetic labels achieve competitive performance (F1: 84.6-87.0 vs 88.3-89.8 for original labels), demonstrating a viable path for extending to tasks without explicit answer locations. Additionally, for larger models (14B+), our self-prompting approach (§2.3, L136-142) *eliminates dependence on labeled data and classifiers entirely*, achieving F1 scores of 78.3-83.1 (Table 1). The emergence of self-assessment capabilities suggests the potential of our method to handle more complex tasks with a training-free alternative (§2.3).
> > > > >
> > > > > **For subjective tasks:** We explore extensions beyond QA tasks in our paper. In Appendix G, we present a preliminary investigation *using in-context learning* (ICL) as a representative non-QA task. Our ICL experiments reveal that different model sizes exhibit varying context requirements, as the smaller model requires nearly all examples to achieve peak confidence, while the larger model can reach good performance with fewer examples. This demonstrates an interesting insight that sufficiency cutoffs could indeed be a model-specific property (§4.4), supporting the broader applicability of our method.
> > > > >
> > > > > **Scope and contributions:** We believe it is important to evaluate our work within the context of academic research and its contributions. Our paper focuses on establishing the fundamental principle that internal LLM representations encode sufficiency signals, facilitate understanding of how LLMs process context. *This previously unexplored aspect of transformer interpretability has value beyond specific implementations and opens new research directions for context-aware processing based on model internal representations.*
> > > > >
> > > > > In this work, we focus on demonstrating core principles with QA tasks rather than attempting to solve all possible tasks in a single paper. This focused approach allows us to conduct comprehensive experimentation to thoroughly validate our method and establish a solid foundation for future research. *The limitation you identify regarding broader task coverage is important for readers to understand, and we will ensure it is clearly stated in our final version. However, this should not diminish our core contributions - demonstrating that dynamic context cutoff is feasible and effective using model internal representations.*
> > > > >
> > > > > Thank you once again for your continued engagement and thoughtful feedback. We hope these clarifications address your concerns to accurately evaluate our work.
> > > > >
> > > > > Best regards,
> > > > > Authors

---

> > > > > > ### Comment · Reviewer_2h3k · 2025-08-02
> > > > > >
> > > > > > Thanks for your response, I will update the score.

---

> > > > > > > ### Author Response · Authors · 2025-08-02
> > > > > > > **Response to Reviewer 2h3k**
> > > > > > >
> > > > > > > Dear Reviewer 2h3k
> > > > > > >
> > > > > > > Thank you immensely for acknowledging our work and updating the score. We truly appreciate the time and effort you invested in providing constructive and valuable feedback!
> > > > > > >
> > > > > > > Best regards,
> > > > > > >
> > > > > > > Authors

---

### Official Review · Reviewer_xgYF · 2025-06-29

**Clarity:** 3
**Significance:** 3
**Originality:** 4
**Rating:** 5
**Confidence:** 3

**Summary:**

This paper introduces a novel approach called dynamic context cutoff for large language model (LLMs), allowing the LLMs to self-assess when they have processed sufficient information from a context and terminate further processing. This is achieved by using lightweight classifiers to detect “sufficiency signals” inherent in certain attention heads. Larger models can also perform this assessment using self-prompting. Experiments demonstrate that this approach can achieve better accuracy and balanced token consumption across various QA tasks, models, and token budget, outperforming current statistic compression and retrieve-based schemes.

**Questions:**

- As the classification threshold is important for the performance, is there a way to automatically set this value for different models, rather than tuning it manually on the validation set?
- Any solution when facing the early cutoff?
- For open-ended, subjective, or tasks lacking clear ground truth positions, how can sufficiency signals be automated? Can synthetic labels be extended to more complex scenarios?

**Ethical Concerns:**

["NO or VERY MINOR ethics concerns only"]

**Final Justification:**

I have read the reviews, participated in the discussions, and provided final feedback.

**Limitations:**

yes

**Paper Formatting Concerns:**

200 Appendix Appendix H.1. -> Appendix H.1.

**Quality:**

4

**Strengths And Weaknesses:**

Strengths:
- The dynamic cutoff method leverages internal model understanding for context truncation, moving away from fixed compression target.
- By identifying sufficiency signals within attention heads, this paper adds evidence to the interpretability of Transformer architecture.
- Achieves significant token savings and accuracy enhancements

Weaknesses:
- Potential early cutoff when the necessary information for answering spans across different parts of the content (like the beginning and end parts).
- This approach depends on the availability of gold answer spans or synthetic sufficiency label generated by another LLM, which could limit the applicability to open-ended, subjective or real-world tasks.

---

> ### Author Rebuttal · Authors · 2025-07-30
>
> Dear Reviewer xgYF,
>
>
> Thank you for your thoughtful and constructive feedback. We appreciate your recognition of our method's strengths, including eliminating the need for pre-defined compression targets, insights into Transformer interpretability, and significant performance improvements. We would like to address your concerns and questions below:
>
> ---
>
> > **Early cutoff when information spans across different parts of content & Solutions for early cutoff**
>
> You raise an important concern about early cutoff. We would like to highlight that our method is carefully designed to handle this scenario. Our dataset construction ensures gold information spans *uniformly* across contexts (Appendix A.2), specifically testing scenarios where information is distributed across different parts. Our multi-hop evaluation on HotpotQA, MUSIQUE, and Multi-hop Key-Value Retrieval requires integrating information from *multiple locations throughout the context*. Table 2 shows our method achieves 29% accuracy on multi-hop tasks, outperforming LLMLingua2 (27.8%) with 1.33× token reduction, demonstrating effectiveness when information spans across distant locations.
>
> Additionally, we address early cutoff through multiple robust mechanisms: (1) High-precision operation: Our method operates at 90% precision with Recall@90P showing 79.3-90.1% coverage, minimizing false positives that cause premature stopping; (2) Ensemble robustness: Using top 4 of 8 trained classifiers provides robustness against individual classifier errors; (3) Adaptive thresholds: Lower thresholds increase recall at efficiency cost, allowing flexible conservativeness-efficiency trade-offs; (4) Fallback guarantee: If no chunk reaches sufficient confidence, we process the entire context, ensuring no information loss in edge cases.
>
> ---
>
> > **Dependence on gold answer spans limiting applicability to open-ended tasks & Extension of synthetic labels to more complex scenarios**
>
> Thank you for the excellent question! Applying our method to scenarios without explicit answer locations is indeed a relevant research direction. We attempted to provide several solutions. In Appendix A.4, we conduct experiments on synthetic labels and find that GPT-4o-generated synthetic labels achieve competitive performance (F1: 84.6-87.0 vs 88.3-89.8 for original labels), enabling extension beyond factoid QA. For larger models (14B+), our self-prompting approach (L180-185 in §2.3) eliminates dependence on labeled data entirely, with Table 1 showing F1 scores of 78.3-83.1. The emergence of self-assessment capabilities also suggests the potential of our method to handle more complex scenarios with a training-free alternative.
>
> For subjective tasks, we believe that the model's internal representations still encode when it has gathered sufficient information to form a coherent response, even if that information is distributed across the document or the answer is not explicitly located in the context. The key point is that *sufficiency is ultimately a property of the model's understanding, not just the dataset structure*. We left this question as a direction for further work but discussed some preliminary findings and ideas under an in-context-learning setting (Appendix G), which does not have explicit answers located in the context.
>
> ---
>
> > **Automatic threshold selection for different models**
>
> This excellent suggestion addresses a key practical deployment challenge. Currently, threshold τ is selected on validation sets per model following standard hyperparameter tuning practices. However, the monotonic confidence progression (Figure 6) suggests potential for automatic threshold adaptation based on confidence curves, which is a valuable direction we acknowledge for future work. Thank you for highlighting this practical insight that could enhance deployment scalability of our method!
>
> ---
>
> Thank you once again for your thoughtful engagement with our work. We appreciate your constructive feedback and look forward to incorporating these insights to further enhance our work.
>
> Best regards,
>
> The Authors

---

> ### Comment · Reviewer_xgYF · 2025-08-02
> **ok**
>
> Thank you for your responses that are both clear and concise. It believe this work should be accepted therefore I keep my score.

---

> ### Author Response · Authors · 2025-08-02
> **Response to Reviewer xgYF**
>
> Dear Reviewer xgYF,
>
> Thank you immensely for the positive assessment of our work! We really appreciate the time and effort you invested in reviewing our work and providing valuable feedback!
>
> Best regards,
>
> Authors

---

### Official Review · Reviewer_6R5V · 2025-06-30

**Clarity:** 4
**Significance:** 4
**Originality:** 3
**Rating:** 5
**Confidence:** 4

**Summary:**

The paper aims to make LLM inference more efficient, on specialized tasks, by teasing out sufficiency signals from the model internals itself. This is mostly relevant for tasks which “find” information in a longer context document, where the model does not have to process the whole text to “find the answer”. Previous work like LLMLingua and RAG methods, always pre-decides a static compression/top-k factor whereas this method always relies on the internal model state to reflect a more dynamic per-context decision of what is sufficient.

Making use of latent knowledge in model activations, is a useful tool that is heavily used in many areas today. This paper attempts to apply this tool for teasing out sufficiency signals to make task-specific inference more efficient and accurate.

The basic idea of the technique is to train a classifier against an appropriately crafted dataset with sufficiency labels, from the internal model state to detect sufficiency signals on the input for the task. This introduces techniques such as chunking where a decision has to be made on how to balance additional inference cost vs accuracy and compression.

The following studies were also done.
- Using self-prompting to detect sufficiency on models of varying sizes.
- Studies on effectiveness signals from attention heads through different layers, on sufficiency signals.
- Comparison of this method against RAG and Lingua2
- Comparison of classifier based methods against supervised fine-tuning and self-prompting methods.
- Efficiency vs Performance trade-off studies  on all methods across multiple models.
- Performance comparisons across multiple models, on different hop characteristics, on short form and long form data.
- Comprehensive classification threshold analysis
- Studies of performance vs cost across different chunking strategies.
- Latency vs accuracy studies

**Questions:**

- Has there been any analysis done on, if the tasks for the prompts need early stopping. For some prompts, it may not need early stopping. Any studies done on detecting whether such early stopping is needed at all? For example, if we extrapolate this to tasks where questions come at the end of the contextual information, can we apply this somehow?
- The authors call the methods of LLMLingua and RAG approach as "static" since they pre-decide some hyper parameters such as compression factor and top-k documents. But in the approach in the paper also, you fix some parameters such as type of chunking and threshold value for the classifiers, which also feel quite similar. How would you clarify this as more dynamic than other static parameters? Can't this be considered as static choices too, especially comparing to the LLMLingua approach? How would you clarify this?
- Since the underlying model already knows the sufficiency signals, especially as the models get bigger, has there been any studies done to let the model itself predict sufficiency using a special token after every chunk? And hence can we get rid of the separate classifier to do the task? Has such a line of thought been explored?
- Is there a chart about the distribution of sufficiency chunk index across multiple datasets? If there is some bias in the distribution of sufficiency signals in the data? Any analysis done on this?

**Ethical Concerns:**

["NO or VERY MINOR ethics concerns only"]

**Final Justification:**

I think this is a strong paper. I maintain my original ratings. The authors have satisfactorily addressed most of the questions raised and have provided clear additional details.

**Limitations:**

Yes. The authors have discussed about the limitations in universal cut-off in sufficiency signals. Further in-depth analysis on it are left to future work.

**Paper Formatting Concerns:**

Line 610: Badly formed sentence.
"Future research should could on designing algorithms that can learn these individualized thresholds"

**Quality:**

4

**Strengths And Weaknesses:**

Overall the paper feels quite strong in all areas.

- The paper is very clear about the problem, methodology, background, experiment, results and interpretation.

- **Strong Arguments**: The paper very clearly presents the arguments to make the case here. It talks about the background of other approaches like LLM Lingua and RAG. It classifies them as static approaches. The authors clearly explain the novelty in the approach in the paper, as making use of model internal state as the differentiating factor. The F1 score of classifiers as a successful metric in distinguishing sufficiency signals are presented as proof that the model internals are sufficient for the purpose. The problem formulation has been well presented in a technically rigorous way and explains the notion of chunks well. Also defines the notion of sufficiency very clearly. A good amount of studies have been done to present a comprehensive argument towards the effectiveness of the method.
- **Clear Presentation**: The approach, data, evaluation benchmarks, comparison data etc have been very clearly mentioned in the paper. Each concept is well articulated in its own section, with the right level of detail. The choice of models to run the experiments on are also appropriate. The baseline comparisons of Fine-Tuned Classifier and Self-Prompting are appropriate baselines to compare against this method.
- **Strong Results**: The results attained by the authors seem to show that with this technique they improve on previous methods on both accuracy and token reduction dimensions, which is decently good. The gap against the baselines seems significant enough to consider this method to be quite effective at token reduction without sacrificing accuracy.
- **Variety of Relevant Studies**: A lot of trade-offs and peripheral impact studies have been explored in the paper which make the results look pretty robust. The list of results along with a combination of dimensions of multiple models, methods, Muti/single hop data types along with comparison to the current methodology gives a good understanding of improvement over the whole landscape of such experiments. In most cases, the method performs the best or close to the best with added advantages. This is presented for short form and long form datasets.
- **Lots of low level details in Code and Appendix**: The presence of a lot of additional prompts, figures and methodologies make it very clear what’s been done. Also appreciate sharing the code as supplementary material.
- The confidence progression chart gives a pretty good confidence that this is indeed doing the right thing. The confidence improves as you add more relevant chunks, which is a good indication.

Overall I feel this builds a good end to end story on the effectiveness of this technique on token reduction without sacrificing accuracy. There are no gaps which are unanswered.

I don’t see any major weakness in the setup, methodology, presentation or arguments.

I would score the paper as 4 on Quality, Clarity and Significance and 3 on Originality.

---

> ### Author Rebuttal · Authors · 2025-07-30
>
> Dear Reviewer 6R5V,
>
> We sincerely thank you for your thorough and constructive review. We are grateful for your positive assessment of our paper's strengths, including the clear problem presentation, strong arguments, comprehensive experiments, and robust results. We especially appreciate your recognition that our work builds “a good end to end story on the effectiveness of this technique” and that you found our approach technically rigorous with no major weaknesses.
>
> We would like to address your questions as follows:
>
> ---
>
>
> > **Has there been any analysis done on whether tasks actually need early stopping? What about tasks where questions come at the end of the contextual information?**
>
> Thank you for this insightful question. We acknowledge that there are tasks that may not benefit from early stopping (eg: passage rewriting, summarization). However, this study focuses on tasks that would benefit from early stopping, which represents a substantial portion of real-world applications (eg: question answering, information retrieval, or fact verification).
>
> A key advantage of our method is the potential to handle both scenarios naturally. Unlike other compression methods that compress regardless of the task, our approach can process the full context when necessary. When all information is crucial for a task, our sufficiency classifier would not trigger early stopping, effectively using the entire input.
>
> Regarding tasks where questions appear at the end of the contextual information: our method is designed to detect when sufficient information has been processed regardless of where the key information appears. In principle, this can be enabled by mixing in training data where we know the full context is needed and training the classifier accordingly. Analyzing whether tasks need early stopping and detecting them explicitly would be a valuable future research direction that could further enhance the applicability of our approach - thank you for pointing out this interesting direction!
>
> ---
>
> > **How is this more “dynamic” than other methods when you still fix parameters like chunking type and threshold value?**
>
> Thank you for this excellent question! We acknowledge this similarity and appreciate the opportunity to clarify our terminology. The key distinction lies in the compression behavior, not the parameter setting approach. In static methods like LLMLingua, a fixed compression ratio is applied uniformly, where a compression ratio of 50% results every input gets compressed to exactly half its original length regardless of content. In contrast, our method uses static hyperparameters to enable content-adaptive compression decisions. Some inputs may receive minimal compression (eg: 5%) while others may be heavily compressed (eg: 95%), with the actual compression amount determined by each input's specific information density and complexity.
>
> The “dynamicity” we refer to is that each input receives different treatment based on its content, whereas other methods compress every input to the same predetermined level. The compression ratios we report are averages across all examples, not fixed targets applied uniformly. We acknowledge that “dynamic” may be misleading terminology and would be happy to use “content-adaptive” or “input-specific” to better describe this distinction.
>
> ---
>
> > **Since larger models already know sufficiency signals, has there been study to let the model predict sufficiency using special tokens instead of separate classifiers?**
>
> Thank you for this excellent observation! The self-prompting (L136-142) method shows the scaling phenomenon as you mentioned. However, even for large models, the probing approach consistently outperforms self-prompting (91.1 vs. 83.1 F1 for 70B model in Table 1). This suggests that while models develop some intrinsic sufficiency awareness, their internal representations contain richer signals than what they can explicitly verbalize.
>
> The special token approach you suggest is intriguing and represents a natural extension. Our probing results (Figure 2) demonstrate that sufficiency signals are most concentrated in specific attention heads of middle layers, which could guide the design of such special tokens. We view this as promising future work that could eliminate the lightweight classifier overhead entirely.
>
> ---
>
> > **Is there analysis of the distribution of sufficiency chunk index across datasets? Any bias in the distribution of sufficiency signals?**
>
> Thank you for this important question! Our datasets are carefully balanced by design (Appendix A.1, Table 7-8), where the gold locations follow uniform distributions across all datasets (mean ≈ 0.50, with standard deviations of 0.25-0.28). This balanced design ensures that ~50% of chunks are classified as “insufficient” and ~50% as “sufficient”. The uniform distribution was intentionally created to provide fair evaluation across different information positioning scenarios.
>
> Our confidence progression analysis (Figure 4) shows that the model's sufficiency confidence increases monotonically as more relevant information is processed, regardless of where that information appears in the context. This validates that our approach can handle diverse information distributions rather than simply learning positional biases.
>
> We also tested our method on naturally occurring datasets where information placement wasn't controlled (see Appendix A.3 on synthetic labels), demonstrating robustness beyond balanced evaluation setup.
>
> ---
>
> We appreciate your thoughtful engagement with our work. Your questions and feedback have allowed us to clarify key aspects of our approach and have highlighted important directions for future research. Thank you once again for your valuable feedback!
>
>
> Best regards,
>
> The Authors

---

> > ### Comment · Reviewer_6R5V · 2025-08-04
> >
> > Thanks for the clarification and useful pointers. Appreciate it.

---

### Official Review · Reviewer_r53E · 2025-07-02

**Clarity:** 3
**Significance:** 3
**Originality:** 3
**Rating:** 4
**Confidence:** 4

**Summary:**

This paper introduces dynamic context cutoff to enable LLMs to self-identify when they have acquired sufficient information for a task, like the human cognitive system. Through analyzing the model internals, the author found that specific attention heads present strong sensitivity to the information sufficiency. By monitoring these attention heads with classifiers, this paper enables LLMs to stop early with sufficient information to save computations. Experiments with static methods, including compression-based approaches and RAG, demonstrate its superiority in both accuracy and token reduction.

**Questions:**

**I have following questions:**

1. Why not chunk input tokens with overlaps?
2. In the introduction section, the author raises one question mentioning ‘teaching’ LLMs which does not appear to align with the paper’s actual methodology. The use of this term may cause confusion.

**Ethical Concerns:**

["NO or VERY MINOR ethics concerns only"]

**Limitations:**

Limitations are pointed out in the paper by the authors.

**Paper Formatting Concerns:**

There are no major formatting issues in this paper.

**Quality:**

3

**Strengths And Weaknesses:**

**Strengths:**

1. This paper introduces a new paradigm for model context reduction, where the capability arises naturally from the model’s own understanding rather than from external compression heuristics, which is both interesting and valuable.
2. The experiments are sound and comprehensive.
3. It is interesting to observe emergent self-awareness of sufficiency through prompting in models with 14B+ parameters.

**Weakness:**

1. As highlighted in the abstract, LLMs are inefficient when the information needed to answer a query is localized within a specific part of the context. Left-to-right cumulative processing remains inadequate in such cases.
2. I have a concern regarding the dataset used to probe LLMs' understanding of context sufficiency in Section 2.2. For instance, if the answer-containing chunk consistently appears near the beginning (left) of the input, it may lead to artificially high classification accuracy without truly demonstrating that specific attention heads are effective indicators of context sufficiency. It would be helpful to report the distribution of the answer-containing chunk index across the dataset. Additionally, more details on the total number of tokens per example should be provided. Another related question is: does reducing the chunk size negatively impact classification performance?
3. More analysis about information loss in this methodology should be provided.

---

> ### Author Rebuttal · Authors · 2025-07-30
>
> Dear Reviewer r53E,
>
>
> Thank you for your insightful reviews and comments! We appreciate your recognition of the novel paradigm and comprehensive evaluation of our work. We would like to address your concerns and questions as follows:
>
> ---
>
>
> > **Dataset Bias and Data Distribution**
>
> You raise an important concern about potential bias for data distribution. Due to page limits, we had to move the dataset details to Appendix. However, we carefully designed our evaluation to avoid this bias. As detailed in Appendix A.1 (Table 6, 7), the gold location (answer position) follows a *uniform distribution* across all datasets:
> - Mean gold location: 0.49-0.51 (middle of context)
> - Standard deviation: 0.25-0.28 (wide spread)
> - Range: 0.01-0.99 (covers entire context span)
>
> This uniform distribution ensures approximately *50% of chunks are insufficient* (before answer) and *50% are sufficient* (after answer), creating a balanced evaluation that prevents the bias you mentioned. We also discuss dataset balance in Appendix A.2 and the collection process in Appendix A.3 in detail.
>
> Regarding token counts: Our contexts range from 0.5K-40K tokens (mean ~23K for long datasets, ~2K for short), with detailed statistics also in Tables 6-7 of Appendix A.1.
>
> ---
>
>
> > **Chunk Size Impact on Classification**
>
> Thank you for this important question! To clarify, our chunks are cumulative and maintained through KV cache reuse. When we process a new chunk, it contains *all previous chunks plus the new segment*, not just the isolated segment alone. This is a crucial distinction that ensures we preserve all contextual information throughout the processing.
>
> Table 4 in Section 4.2 demonstrates that smaller chunks actually improve classification performance, with sentence-level chunking achieving F1=96.8 compared to 88.3 for 10% chunking. This is because smaller chunks allow for more frequent sufficiency checks, enabling earlier detection of relevant information. However, smaller chunks increase computational overhead due to more frequent forward passes. Therefore, we chose 10% chunking as the optimal balance between accuracy and efficiency.
>
> ---
>
>
>
> > **Left-to-Right Processing**
>
> Thank you for this excellent question. While left-to-right processing may limit efficiency in some cases, our method offers a couple other advantages. Due to page limits, we had to move the discussion on the processing order to Appendix. However, as discussed in Appendix B.3, our approach offers several advantages compared to other processing orders: it maintains semantic coherence by preserving natural left-to-right text flow, enables KV cache reuse for computational efficiency, and most importantly, this processing order aligns with how current transformer models are trained and operate in practice.
>
> While alternative processing orders could be explored, our current approach still achieves significant improvements with a 3.4% accuracy gain and 1.33X token reduction compared to static methods. We believe investigating optimal processing orders remains an interesting direction for future work, but our current results demonstrate the effectiveness of the left-to-right approach.
>
> ---
>
>
> > **Information Loss Analysis**
>
>
> Thank you for this excellent suggestion! To comprehensively understand the tradeoffs between information loss and performance in our methodology, we would like to highlight the analysis in Sections 4.1 (Classification Threshold) and 4.2 (Chunking Strategy). However, the most compelling evidence that our method preserves rather than loses critical information is the consistent performance improvement. Specifically, we see a 3.4% average accuracy improvement (Tables 2-3), indicating our method helps models focus on relevant content rather than discarding important information.
>
> To address concerns about false positives that could potentially lead to information loss, we report Recall at Precision 90-98% (Figures 4, 10, 11). Our threshold mechanism allows trading off precision for efficiency, where higher precision settings ensure no information loss at the cost of lower efficiency. Additionally, Figure 6 shows that models accumulate useful signals progressively, suggesting our approach captures relevant information effectively rather than making premature cutoff decisions.
>
>
> ---
>
>
> > **Why not chunk input tokens with overlaps?**
>
> Thank you for this important question! We would like to clarify that each chunk extends the context incrementally (chunk₁, chunk₁+₂, chunk₁+₂+₃, etc.), which allows us to reuse the KV cache and avoid redundant computation of previously processed tokens. This cumulative approach maintains semantic continuity by preserving natural text flow without creating artificial boundaries that could disrupt understanding. On the other hand, overlapping chunks would require recomputing activations for the same tokens multiple times, which would negate our computational efficiency gains while disrupting the cumulative context semantics. The non-overlapping design ensures that each new chunk only requires computing activations for the truly new tokens, maximizing both computational efficiency and contextual coherence.
>
>
> ---
>
>
> > **”Teaching” Terminology Confusion**
>
> This is an excellent point! Our method doesn't “teach” LLMs new capabilities but rather leverages existing internal representations. We will use “enabling” or “allowing” to avoid confusion.
>
> ---
>
> We appreciate your suggestions and will incorporate them to enhance our work further. Thank you once again for your valuable feedback!
>
>
> Best regards,
>
> The Authors

---

> > ### Comment · Area_Chair_ryMe · 2025-08-05
> > **Reminder: Reviewer r53E first response**
> >
> > Dear reviewer r53E,
> >
> > This is a reminder to post your first response, as the deadline of author-reviewer discussion period is closing. The authors have responded to your reviews, and also to the others reviews. Please discuss openly with the authors, regarding your reviews and the addressed questions from the authors.

---

> > > ### Comment · Reviewer_r53E · 2025-08-06
> > >
> > > Thank you for the clarifications. I believe my concerns have been addressed

---

### Comment · Area_Chair_ryMe · 2025-08-05
**Reminder: end of author-reviewer discussion period**

Dear reviewers,

This is a reminder that the end of author-reviewer discussion period is near. Please do carefully read all other reviews and the author responses; and discuss openly with the authors, especially on your own questions that the authors addressed.

Thank you.

---

### Note · Authors · 2025-08-15

We are happy to see the very positive comments by the reviewers and grateful for the constructive feedback.

The reviewers highlight that our work is successful and important without any critical flaws. We also appreciate the positive comments on our novel method (r53E, 6R5V, xgYF), comprehensive experimentation (r53E, 6R5V, 2h3k), and strong results (r53E, 6R5V, xgYF). All concerns raised by reviewers are fully addressed and discussed during the discussion period. The main suggestions include improving the presentation by moving the discussion of other efficiency techniques and discussion of the generalization across different task types from the appendix to the main text. We are committed to doing so upon acceptance of the paper, and we think that this way the important insights of the study will be covered in the main paper (and not in the appendices).

We are also committed to addressing all requested changes and incorporating suggestions from the discussion with reviewers before the camera-ready – in fact, we have already completed most of these changes, as none of them required additional experiments. Thank you for your consideration.

---

### Decision · Program_Chairs · 2025-09-17

**Decision:**

Accept (poster)

**Comment:**

This paper introduces dynamic context cutoff for LLMs to self-identify sufficient context in processing longer context document, where usually the information required is localized. The authors analyze model internals and found that specific attention heads inherently encode such sufficiency signals accessible through lightweight probing classifier. Furthermore, larger models also exhibit this intrinsic self-assessment through prompting. The author showcase that this intrinsic self-assessment capability enables LLMs to stop early with sufficient information, and in turn, demonstrates token reduction without sacrificing accuracy—or even improving it—across various QA tasks and models, outperforming current compression-based and retrieval-based methods.

__Strengths__
- The introduction of new paradigm for intrinsic model context reduction is both interesting and valuable
- The writing of this paper is very clear
- The experiments are abundant, sound, and comprehensive
- Observation of emergent self-awareness of sufficiency through prompting in larger models is interesting

__Weakness__
- The rebuttals addressed well on the concern of technical innovation, probing dataset, information loss, cutoff failures, hyperparameter sensitivity, and generalization towards broader real-world tasks.

The empirical gains, breadth of validation, and mechanistic insights outweigh concerns about generality and the simplicity of the probing classifier. The rebuttals well addressed the concern of technical innovation, probing dataset, information loss, cutoff failures, hyperparameter sensitivity, and generalization towards broader real-world tasks. In conclusion, I believe the paper presents an interesting and meaningful contribution to the community.